# Lifestyle and Biochemical Parameters That May Hamper Immune Responses in Pediatric Patients After Immunization with the BNT162b2 mRNA COVID-19 Vaccine

**DOI:** 10.3390/diseases13030078

**Published:** 2025-03-10

**Authors:** Anthie Damianaki, Antonios Marmarinos, Margaritis Avgeris, Dimitrios Gourgiotis, Elpis-Athina Vlachopapadopoulou, Marietta Charakida, Maria Tsolia, Lydia Kossiva

**Affiliations:** 1Second Department of Pediatrics, School of Medicine, National and Kapodistrian University of Athens, Children’s Hospital P. and A. Kyriakou, 115 27 Athens, Greece; charakidadoc@googlemail.com (M.C.); mariantsolia@gmail.com (M.T.); lydiakossiva@hotmail.com (L.K.); 2Laboratory of Clinical Biochemistry—Molecular Diagnostics, Second Department of Pediatrics, School of Medicine, National and Kapodistrian University of Athens, Children’s Hospital P. and A. Kyriakou, 115 27 Athens, Greece; antmar@med.uoa.gr (A.M.); mavgeris@med.uoa.gr (M.A.); dgourg@med.uoa.gr (D.G.); 3Department of Endocrinology-Growth and Development, Children’s Hospital P. and A. Kyriakou, 115 27 Athens, Greece; elpis.vl@gmail.com

**Keywords:** obesity, BMI, humoral immunity, uric acid, insulin resistance, mRNA vaccine

## Abstract

Background: The aim of this study was to evaluate whether increased body mass index (BMI) and biochemical and lifestyle parameters linked to obesity and smoke exposure disrupt immune responses of children and adolescents following vaccination with the mRNA BNT162b2 vaccine. Methods: A prospective, single-center, cohort study was conducted. Participants were assigned to receive two doses of the mRNA vaccine. Anti-SARS-CoV-2 IgG and neutralizing antibodies (AB) were measured before vaccination (T0) and 14 days after the second dose (T1). BMI and biochemical parameters were evaluated at T0. A questionnaire on lifestyle characteristics was filled in. Results: IgG optical density (OD) ratio at T1 was lower in the overweight–obese group regardless of COVID-19 disease positive history [*p* = 0.028 for the seronegative group, *p* = 0.032 for the seropositive group]. Neutralizing AB were lower in overweight–obese participants in the seronegative group at T1 [*p* = 0.008]. HDL, fasting glucose/insulin ratio (FGIR), C-reactive protein (CRP), HBA1c, uric acid, and smoke exposure were significantly correlated with BMI [*p* = 0.006, *p* < 0.001, *p* < 0.001, *p* = 0.006, *p* = 0.009, *p* < 0.001, respectively]. The main biochemical parameters that were inversely correlated with IgG and neutralizing AB titers at T1 were uric acid [*p* = 0.018, *p* = 0.002], FGIR [*p* = 0.001, *p* = 0.008] and HBA1C [*p* = 0.027, *p* = 0.038], while smoke exposure negatively affected the humoral immune responses at T0 in the convalescent group [*p* = 0.004, *p* = 0.005]. Conclusions: Current data suggests that uric acid, insulin resistance (IR), and smoke exposure could adversely affect the immune responses in overweight–obese vaccinated children, highlighting the need for actions to enhance the protection of this particular subgroup.

## 1. Introduction

Severe acute respiratory syndrome coronavirus-2 (SARS-CoV-2) emerged in Wuhan in December 2019 and rapidly escalated into a global health emergency, causing significant social and economic disruptions worldwide. Active immunization appeared to be the cornerstone of healthcare policies globally in order to reach herd immunity for long-term control of the SARS-CoV-2 pandemic [1]. Current studies demonstrate how COVID-19 vaccines have resulted in robust humoral responses in the pediatric population up to 1 year following immunization, while vaccination against SARS-CoV-2 was linked to a lower recurrence infection rate and showed a trend toward reducing the prevalence of other flu-like illnesses [2,3].

Obesity has been identified as a major risk factor for severe illness, hospitalization, and mortality during the COVID-19 pandemic in both adult and pediatric populations. It is associated with a state of chronic, low-grade inflammation that leads to impaired immunity and to defective humoral responses after vaccination, rendering obese people more susceptible to infections [4]. Obesity is an ongoing global health issue in many countries, including Greece. According to the Organization for Economic Cooperation and Development (OECD) records, the prevalence of obesity in Greece is higher compared to that in other countries of OECD, affecting 17% of the population [5]. It is estimated that 41% of Greek children aged 5 to 9 years are diagnosed with overweight or obesity whereas the average percentage in other OECD countries is 31.4% [6].

The aim of the current study was to analyze the COVID-19 vaccine response in correlation to BMI and how biochemical parameters such as inflammation markers, lipidemic profile, IR, and lifestyle parameters such as smoke exposure accompanying obesity can play a major role in immunogenicity of children and adolescents.

## 2. Materials and Methods

### 2.1. Study Population and Design

A prospective, single-center, cohort study of Greek children and adolescents who received the COVID-19 vaccine (mRNA BNT162b2) from October 2021 to October 2022 at the P. and A. Kyriakou Athens Children’s Hospital was conducted. The study population’s clinical and epidemiological data and laboratory and immunological indices were examined.

The enrolled vaccinated subjects met the following inclusion criteria: (1) age between 5 and 18 years old, (2) already assigned to receive the mRNA BNT162b2 vaccine during the National Immunization Program against SARS-CoV-2, (3) provided written informed consent from their legal guardians, (4) agreed to two blood samplings at T0 and T1. Subjects with a history of current or recent febrile illness, frequent or severe respiratory illnesses, history of recent vaccination, immunosuppression-associated underlying disease or under treatment with immunosuppressive drugs, as well as participants with growth hormone deficiency, Cushing syndrome or syndromic cases were excluded from the study.

A questionnaire was used to retrieve data on the participants’ sociodemographic and healthcare characteristics. The data were collected via face-to-face interviews by qualified personnel. Past COVID-19 disease history was confirmed by positive polymerase chain reaction (PCR) and/or rapid diagnostic test (RDT). Lifestyle parameters including dietary and sleep habits, physical activity, smoke exposure and/or smoking or vaping, and alcohol use were recorded. Smoking was defined by active tobacco use by the participants and vaping, which is more popular among adolescents, was assessed by the use of e-cigarettes. Smoke exposure was considered significant when their guardians were active indoor smokers with more than 20 pack years [7]. Dietary lifestyle was assessed by KIDMED score reflecting the grade of adherence to the Mediterranean diet (poor: 0–3, moderate: 4–7, good: ≥8) [8].

A binary score was used for smoke exposure and/or smoking or vaping, alcohol use, exercise, and sleep habits to produce an adequate sample size for the pattern analysis. The participants received 1 point for each factor if they were nonsmokers or not exposed to secondhand smoke by their guardians, did not consume alcohol, performed regular physical activity (>3 or 4 times per week), and had a normal bedtime sleep schedule according to age (>9 h for the participants aged from 5 to 11 years and >8 h for those aged from 12 to 18 years); otherwise, they received 0 points for each corresponding factor [9,10].

BMI, defined as weight (kg)/height^2^ (m^2^), was measured in each participant at T0 and the participants were categorized as overweight or obese when weight was above the 85th or the 95th percentile for age and gender according to the CDC (Center of Disease Control) curves [11]. Blood pressure (BP) was also measured with the appropriate cuff.

All eligible participants were assigned to receive two doses of the mRNA vaccine BNT162b2 during the National Immunization Program against SARS-CoV-2.

The manufacturers’ (BioNTech/Pfizer, BioNTech Manufacturing GmbH, Mainz, Germany) directives for storage and administration were followed. In accordance with the CDC and local guidelines, all participants aged 12 years and above received two doses of 30 μg (0.3 mL each) and all children aged 5 to 11 years received two doses of 10 μg (0.2 mL each), 3 to 4 weeks apart given intramuscularly into the deltoid muscle.

Participants were stratified according to BMI and COVID-19 disease history prior to immunization.

### 2.2. Assessment of SARS-CoV-2 Binding Antibody and Biochemical Parameters

The protocol included anti-SARS-CoV-2 anti-spike IgG and neutralizing AB measurement prior to vaccination (T0) and 14 days after the second dose (T1).

SARS-CoV-2 neutralizing AB were assessed using the cPass SARS-CoV-2 Neutralization Antibody Detection Kit (GenScript Biotech Corp., Piscataway, NJ, USA), employing a blocking enzyme-linked immunosorbent assay (ELISA) approach. The assay exhibits intra-assay and inter-assay variabilities of 8% and 10%, respectively, with a detection limit of 2%. For SARS-CoV-2 anti-spike IgG measurement, we used the SERION ELISA agile SARS-CoV-2 IgG kit (Institut Virion/Serion GmbH, Wurzburg, Germany) based on the standard ELISA principle. The assay demonstrates intra-assay and inter-assay variabilities of 2.3% and 3.5%, respectively, with a detection limit of 1 U/mL [12,13].

Due to enormously high values of anti-spike IgG AB impeding a quantitative measurement according to the standard use of the kit (maximum quantitative measurement 5000 U/l), we evaluated the immunogenicity of the cases using the OD values based on the following cutoffs: <0.25, negative; ≥0.25 to <0.4, borderline; and ≥0.4 positive. OD values were measured at 405 nm using the Tecan Infinite M200 reader (Tecan Group Ltd., Zürich, Switzerland).

The blood samples were centrifuged at 1600× *g*, at 4 °C for 14 min and the serum supernatant was stored at −80 °C. All serum samples were analyzed in the same batch at an ambient temperature of 24 °C by experienced personnel who were blinded to the individual’s group (overweight–obese or normal-weight).

Biochemical parameters including CRP, uric acid, lipidemic profile [total cholesterol (TCHOL), low-density lipoprotein cholesterol (LDL-C), high-density lipoprotein cholesterol (HDL-C), triglycerides (TGs), lipoprotein a: Lp (a)] and IR parameters such as insulin, glucose, and HBA1c, were also surveyed from morning fasting blood samples at T0 [14,15].

Concerning lipidemic profile, CRP and uric acid, blood samples were spun at 1600× *g*, at 20 °C for 7 min. The photometric method was used to analyze the serum samples via Cobas c501 analyzer (Roche diagnostics) and an immunoluminometric assay (ILMA) was used for measuring insulin via the Cobas e601 analyzer (Roche diagnostics); spinning 1600× *g* at 20 °C for 10 min preceded the latter.

HbA1c levels in whole blood were measured using the Hb NEXT (Menarini diagnostics), an automated HPLC (high-performance liquid chromatography) analyzer.

Fasting serum glucose (mg/dL) to plasma insulin (micro U/mL) ratio (FGIR) was used as the main surrogate index of IR with a cutoff of 6.5 [16]. Lower levels of FGIR are interpreted as greater IR.

### 2.3. Statistical Analysis

Qualitative variables were expressed as numbers with frequencies and percentages. Continuous variables were expressed as a median (interquartile range: IQR) or mean ranks when the data did not follow a normal distribution and as a mean (95% CI: 95% BioNTech/Pfizer confidence interval) for the normally distributed variables. Student’s T-test was used for paired comparisons and the Mann–Whitney test was used for unpaired comparisons. Kruskal–Wallis was used when comparing multiple groups. Categorical variables were compared using the chi-square (χ^2^) test and Spearman’s rank correlation coefficient was used to assess correlations between continuous quantitative variables. The association between humoral AB titers and biochemical/lifestyle parameters was estimated via a stepwise method of multivariate linear regression both in seropositive and seronegative subgroups. All statistical analyses were conducted using IBM SPSS Statistics version 25.0 for Windows. Statistical significance was defined as a *p*-value < 0.05.

## 3. Results

### 3.1. Results of Humoral Immune Responses of Study Population Divided by BMI and Characteristics of Study Population

One hundred-and-eight (*n* = 108) subjects were enrolled in the study. Twenty-nine (*n* = 29) participants were excluded due to dropout. After the second dose of BNT162b2, all of the vaccinated subjects developed a humoral immune response compared to the baseline (T0) serum level. As depicted in Figure 1, the anti-SARS-CoV-2 IgG ELISA OD ratio at T1 was higher in the group with a positive previous COVID-19 disease history (*p* < 0.001, Figure 1a), while SARS-CoV-2 neutralizing AB levels were almost the same in the two groups (*p* = 0.35, Figure 1b).

Anti-SARS-CoV-2 IgG OD ratio at T1 was significantly lower in the overweight–obese group compared to normal-weight group regardless of pre-vaccination COVID-19 disease history [*p* = 0.028; negative history and *p* = 0.032; positive history, Table 1]. Neutralizing AB levels were lower in overweight–obese participants with statistical significance only in the previously seronegative group [*p* = 0.008]. BMI was significantly associated with immune responses in convalescent sera at T0 regarding both IgG and neutralizing AB titers [*p* = 0.026 and *p* < 0.001].

### 3.2. Characteristics of Study Population

Among the 79 participants, 5 reported an underlying disease in their medical history; 3 mentioned hypothyroidism, 1 had type 1 diabetes (insulin-dependent), and 1 reported epilepsy. They were all under treatment and their underlying disease was under control. All had a normal thyroid profile. Insulin resistance parameters (glucose, insulin, HBA1c) of the individual with type 1 diabetes were ruled out. The lipidemic parameters of one participant were also excluded from the study due to probable undiagnosed dyslipidemia (TCHOL: 62 mg/dL, LDL: 15 mg/dL).

None of the participants in any age group reported sleep disorders, irregular bedtime schedules, or alcohol use. Regarding blood pressure (BP) measurements, only 7 out of 79 participants (5 with BMI > 85th percentile and 2 with BMI <85th percentile) showed BP values between the 90th and 95th percentiles. However, after being referred to their physicians, twice-daily BP measurements over a two-week period revealed normal BP values, indicating a ‘white coat’ phenomenon. Consequently, sleep habits, alcohol consumption, and BP values were excluded from the analysis. All participants were of Greek origin, with 50.6% (40/79) being female and 63.3% (50/79) classified as adolescents (ages 12–18). None of the participants reported active smoking or vaping.

The study population was initially divided into two subgroups based on BMI (Table 2). Forty-five percent (36/79) were categorized as overweight or obese, with an equal distribution of female and male participants. The five participants reporting hypothyroidism, insulin-dependent diabetes, and epilepsy belonged to the normal-weight group. Of the overweight–obese group, 22 out of 36 were aged 12–18 years. A physical examination detected Acanthosis Nigricans and buffalo hump in 10/22 (45%) adolescents with overweight–obesity; 7 out of 10 were male. Overweight–obese patients were 1.64 times more likely to have been exposed to smoke compared to those with normal weight [Χ^2^_(1)_ = 10.67, *p* < 0.001, *Cramer’s V* = 0.4, *p* < 0.001, Table 2].

During the 6 months prior to enrollment, 33/79 participants reported a positive history of mild COVID-19 disease. Mild disease history included afebrile status or low-grade fever (<38.5 °C), nasal congestion, and/or sore throat and was verified by detection of abs at T0. RDT for SARS-CoV-2 was negative at T0. Almost half of the seropositive individuals (16/33) were overweight–obese and 54.5% (18/33) belonged to the younger age group. Younger participants were 2.8 times more likely to have a positive history for COVID-19 disease prior to vaccination [X^2^_(1)_ = 7.76, *p* 0.01, *Cramer’s V* = 0.31, *p* 0.01].

### 3.3. Results of Biochemical Profile of Study Population

Lower levels of HDL and FGIR and higher levels of CRP, HBA1c, and uric acid were documented in the overweight–obese group with statistical significance regardless of pre-vaccination disease history (*p* = 0.006, *p* < 0.001, *p* < 0.001, *p* = 0.006, *p* = 0.009, respectively, Table 2).

When the population was divided according to COVID-19 disease history, CRP and FGIR were significantly higher in the overweight–obese group in both subgroups. HDL, uric acid, and HBA1c were statistically correlated with BMI only in the immune-naive group [Appendix A, Table A1]. COVID-19 disease history was not correlated with the latter biochemical parameters via parametric tests.

### 3.4. Results of Humoral Immune Responses of Study Population According to BMI, Age, and Gender

When the population was divided by BMI and disease history prior to immunization and correlated simultaneously with age and gender, the female vaccinees with overweight–obesity demonstrated significantly higher IgG ab responses both in the convalescent and the immune-naive subgroup than the male vaccinees with overweight–obesity [Figure 2a: IgG OD (CI 95%): 2.79 (2.60–2.85) vs. male IgG OD (CI 95%): 2.41 (2.21–2.61), t(14)= −3.24, *p =* 0.009 and Figure 2b: female IgG OD (CI 95%): 2.54 (2.36–2.62) vs. male IgG OD (CI 95%): 2.19 (2.15–2.53), t(18)= −1.43, *p* = 0.018]. Finally, participants aged 5 to 11 years showed also higher neutralizing ab levels in overweight–obese and seronegative subgroup [Figure 2c: 5–11 years old median neutralizing ab (IQR): 97.78 (2.13) vs. 12–18 years old median neutralizing ab (IQR): 95.76 (1.54), *U=* 10, *p* = 0.037] and in normal-weight and seropositive subgroup [Figure 2d: 5–11 years old median neutralizing ab (IQR): 98.40 (1.54) vs. 12–18 years old median neutralizing ab (IQR): 97.73 (0.95), *U =* 14, *p* = 0.036].

### 3.5. Impact of Smoke Exposure, BMI, FGIR, HBA1c, and Uric Acid on Humoral Immune Responses of Study Population

A stepwise method of multivariate linear regression of anti-SARS-CoV-2 IgG ELISA OD ratio and SARS-CoV-2 neutralizing AB levels for potential confounding was performed by the inclusion of covariates such as BMI, age, gender, biochemical profile (uric acid, TG, FGIR, and HBA1c) and lifestyle parameters (smoke exposure, physical activity, KIDMED score). Data on multivariate analysis of humoral responses at T1 and at T0 for the convalescent group are documented in Table 3. Immune responses of the convalescent group before immunization who were exposed to smoke were 55% lower for anti-SARS-CoV-2 IgG responses (b = −0.655, t = −3.14, *p* = 0.004) and 25.61% lower for SARS-CoV-2 neutralizing AB (b = −25.61, t = −3.04, *p* = 0.005) compared to those who were recovered from COVID-19 disease prior to vaccination but not exposed to smoke. A 1 kg/m^2^ increase in BMI resulted in the IgG ratio falling by 2% (b = −0.02, t = −3.36, *p* = 0.002) and neutralizing AB falling by 0.17% (b = −0.17, t = −4.47, *p* < 0.001) in the immune-naïve group, while a 1 kg/m^2^ increase in BMI led to the IgG ratio decreasing by 3% (b = −0.03, t = −5.17, *p* < 0.001) and neutralizing AB by 0.15% (b = −0.15, t = −2.80, *p* = 0.009) in the convalescent group. Finally, concerning the latter group, a 1% increase in HBA1c led to neutralizing AB decreasing by 1.36% (b = −1.36, t = −2.20, *p* = 0.038) and IgG ratio by 17% (b = −0.17, t = −2.30, *p* = 0.027). If the participants had IR, the percentage of the neutralizing AB and the IgG ratio was 1.52% lower (b = −1.52, t = −2.83, *p* = 0.008) and 24% lower (b = −0.24, t = −3.64, *p* = 0.001), respectively, compared to those who did not have IR. In contrast, uric acid seemed to play a role in the immune-naïve group: a 1 mg/dL increase of uric acid led to the IgG ratio falling by 6% (b = −0.06, t = −2.47, *p* = 0.018) and neutralizing AB by 0.59% (b = −0.59, t = −3.28, *p* = 0.002).

Chi-square tests showed FGIR was statistically correlated with smoke exposure with the possibility of demonstrating IR while exposed to smoke being 2.32 higher than those not exposed to smoke [Χ^2^_(1)_ = 5.77, *p* = 0.016, *Cramer’s V* = 0.27, *p* = 0.017], but not statistically associated with age, gender, physical activity, KIDMED score, and COVID-19 history before vaccination, albeit there was a tendency of the older subgroup to display 1.64 times lower FGIR than the younger ones [Χ^2^_(1)_ = 3.32, *p* = 0.061]. Finally, we conducted a logistic linear regression of FGIR as the dependent categorical variable with BMI, age, gender, smoke exposure, physical activity, KIDMED score, and COVID-19 disease history before immunization to predict if any of the aforementioned variables could increase the possibility of demonstrating IR. Only BMI was statistically correlated with IR and specifically a 1 kg/m^2^ increase of BMI seemed to increase the possibility of IR by 68% [exp(B) = 1.679, B = 0.518, *p* < 0.001, Appendix B, Table A2].

## 4. Discussion

The COVID-19 pandemic has prompted the development of a plethora of vaccines using various platforms. The BNT162b2 COVID-19 vaccine is a mRNA–lipid nanoparticle vaccine encoding the SARS-CoV-2 spike protein (S) stabilized in the prefusion conformation [17,18]. Many studies have shown that this vaccine contributes to a robust protection from severe disease in children and adolescents [19,20]. In our study, we tried to assess how increased BMI and a deranged biochemical profile, as well as lifestyle parameters linked to obesity such as smoke exposure, could have a negative impact on immune responses of children and adolescents following vaccination with the mRNA BNT162b2 vaccine.

Herein, humoral immune responses were compared between the seropositive and the seronegative groups at enrollment since about 40% of participants had a history of COVID-19 disease prior to immunization. The immune responses, especially the anti-SARS-CoV-2 anti-spike IgG levels, elicited in the convalescent group 14 days after the booster dose of BNT162b2 COVID-19 vaccine were significantly higher than in the immune-naive group. This aligns with previous studies indicating that mRNA vaccines trigger rapid humoral responses in seropositive individuals with post-vaccine antibody titers [21,22]. On the other hand, virus neutralization is the main surrogate marker of vaccine robust humoral response against COVID-19 and mRNA vaccines are known to protect against severe disease [23]. When infected by the virus, complement activation increases immune responses by enhancing antibody neutralization of SARS-CoV-2; thus, seropositive and seronegative vaccinated subjects reaching almost the same neutralizing AB levels at T1 could be expected [24,25].

Greece has been identified as one of the southern European countries with the highest rates of childhood obesity [26]. Overweight and obesity are major risk factors for COVID-19 disease in all age groups, underlying the necessity for an effective COVID-19 vaccine in individuals with obesity [27]. Several studies have already shown that obesity is associated with impaired immune responses to several immunizations including those against influenza, hepatitis B, tetanus, and SARS-CoV-2 [28,29,30,31]. Adipose tissue in obesity is characterized by the infiltration of interferon (INF)-γ-producing CD8+ and Th1 CD4+ T cells, which promote the secretion of pro-inflammatory cytokines by macrophages and leads to chronic, ongoing inflammation and contributes to local and systemic IR. By contrast, in the adipose tissue of normal-weight individuals the Th2 and T-regulatory (Tregs) CD4+ cells predominate and promote secretion of IL-10 and other anti-inflammatory cytokines from macrophages [32]. Manna et al. described the dysfunction of the adipose tissue of overweight–obese individuals at the biochemical level, finding that an overload of intracellular and plasma glucose and free fatty acids lead to the generation of superoxide anion (O2•−), hydrogen peroxide (H_2_O_2_), and hydroxyl radicals (OH•) that constitute the reactive oxygenated species (ROS) by activating nicotinamide adenine dinucleotide phosphate (NADPH) oxidases. In obesity, these superoxide radicals are increased compared to antioxidants and are capable of the activation of transcription factors, such as the nuclear factor kB (NF-kB), promoting the production of pro-inflammatory cytokines and chemokines [33,34]. These findings of inflammation in overweight–obese individuals damaging adaptive immunity are compatible with our results described in Table 1 and Table 3. It was shown that increased BMI, elevated HBA1C levels, and low FGIR levels were associated with a reduction of COVID-19 vaccine-induced humoral immunity responses two weeks after the booster dose. In our study, FGIR and HBA1C negatively affected humoral responses only in the seropositive group [Table 3]. Although there are studies supporting that COVID-19 infection often leads to glycemic derangement in patients with or without diabetes, a logistic linear regression model demonstrated that IR was only linked to increased BMI in our population [Appendix B, Table A2].

Obesity has been associated with elevated uric acid, CRP, IR (low FGIR), and lower HDL levels that are in line with our results [Table 2]. The lower HDL levels could also be explained by the higher percentage of normal-weight participants reaching a good exercise score compared to the overweight–obese group, although these differences did not reach a statistically significant level [35]. Uric acid is the end product of purine metabolism in humans. Studies have demonstrated that the mRNA expression and activity of xanthine oxidoreductase is increased in the adipose tissue of individuals with obesity, leading to higher production and secretion of uric acid, whereas increased IR has been postulated to also increase serum uric acid by decreasing its renal clearance [36,37,38]. Although uric acid possesses an antioxidant capacity, this can be reversed at higher than normal levels. The literature supports that imbalances in micronutrients like uric acid can disrupt defense reactions against diseases or affect immune responses after vaccination [39]. According to Kubota et al., hyperuricemia in children and adolescents is defined as a serum uric acid level exceeding 2 SD above the mean [40,41]; therefore, a threshold of 6.5 mg/dL was set in this study for the abnormal values. Although only 5 participants with obesity had exceeded this threshold, uric acid was inversely correlated with immune responses [Table 3].

Age and gender are two significant demographic factors that play a crucial role in vaccine-induced immunity. Numerous studies have reported that younger individuals tend to exhibit favorable immune responses due to the abundance of switch memory B-cells, less inflammatory cytokines, and the capacity to rapidly produce broadly neutralizing AB [42]. In our study, which included only children and adolescents, the 5–11 year old participants (29 out of 79) exhibited even higher neutralizing antibody titers than their older counterparts, regardless of disease history or BMI. We might hypothesize that this phenomenon is due to the higher incidence of infection by the virus prior to immunization because of the emergence of SARS-CoV-2 variants with high transmissibility (e.g., Omicron). In addition, considering that the two age groups consisted of the same percentage of participants with overweight–obesity and normal-weight participants, this difference in immune response could be attributed to the lower FGIR displayed by the older ones, probably due to the transient insulin sensitivity decrease during puberty, although the literature supports that in otherwise healthy adolescents, pubertal IR is accompanied by compensatory insulin secretion [43]. The female group demonstrated higher humoral responses at T1 even in the overweight–obese subgroup. Previous studies that examined the impact of gender on vaccine responses have shown that females usually mount more robust humoral and cellular immune responses to vaccination and infection, probably due to estrogen receptors (Era/β) being expressed on plenty of immune cells [44].

Finally, smoke exposure was significantly correlated with increased BMI and IR in our study, which is in agreement with previous studies supporting that pediatric obesity and passive smoking are interconnected [45]. Many reports have indicated the negative effect of smoking on both innate and adaptive immunity. Currently active smoking has been associated with immune cell count remodeling, production of inflammatory cytokines that further activate antibody-producing T cells leading to a cascade of reactions in response to the oxidative stress of smoking, and decreased production of immunoglobulins (IgG, IgA, IgM) [46,47,48]. Vardavas et al. reported that passive or second-hand smoking is capable of altering CD4+CD45RA+/CD4+CD45RO+ T-cell circulating subpopulations in the pediatric population, provoking disease predisposition [49]. According to our findings, smoke exposure was negatively correlated with humoral titers of the convalescent group only at T0 [Table 3].

The present study has several limitations that deserve comment. It was a single center study and the number of participants was limited. Several data regarding lifestyle parameters were self-reported; thus, the potential effect of recall bias cannot be ruled out. They were all of Greek (Caucasian) ethnicity, therefore the sample is not representative of the entire population nor of the non-Caucasian population. Overweight and obesity was described by BMI, which cannot differentiate increased body weight from excessive fat-mass or fat-free mass. There was no documentation of steroid hormone levels in the study population. T- cell immunity was not analyzed in the study and smoke exposure was evaluated by parent reports of active indoor smoking and not by measuring a metabolite of nicotine (serum cotinine levels), as reported in other studies [50].

## 5. Conclusions

The presented research aimed to identify the key biochemical and lifestyle parameters that accompany the pediatric population with overweight–obesity and could adversely affect the immune responses to SARS-CoV-2 immunization. Considering that children have a more naive immune system that evolves with age, it is important to mitigate factors that may have a negative impact on the immune response. The adoption of healthy life habits and the development of efficient vaccination programs are of paramount importance.

## Figures and Tables

**Figure 1 diseases-13-00078-f001:**
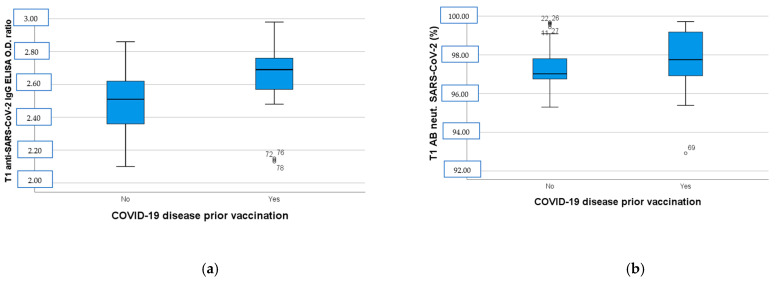
(**a**) Anti-SARS-CoV-2 IgG ELISA OD ratio at T1 in participants with negative and positive COVID-19 disease history before immunization. (**b**) SARS-CoV-2 neutralizing AB levels at T1 in participants with negative and positive COVID-19 disease history before immunization. T1: 14 days after BNT162b2 booster dose, AB: antibodies, OD: optical density.

**Figure 2 diseases-13-00078-f002:**
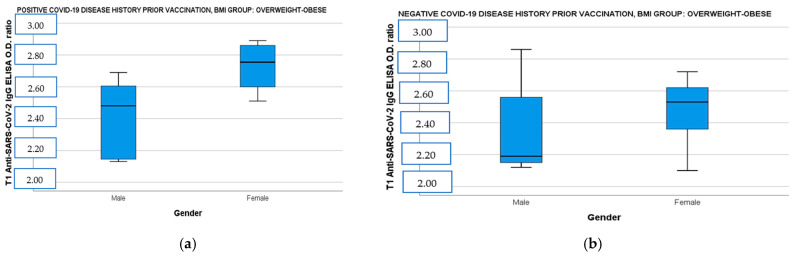
(**a**) Anti-SARS-CoV-2 IgG ELISA OD ratio at T1 in overweight–obese participants with a positive COVID-19 disease history before immunization divided by gender. (**b**) Anti-SARS-CoV-2 IgG ELISA OD ratio at T1 in overweight–obese participants with a negative COVID-19 disease history before immunization divided by gender. (**c**) SARS-CoV-2 neutralizing AB levels at T1 in overweight–obese participants with a negative COVID-19 disease history before immunization divided by age. (**d**) SARS-CoV-2 neutralizing AB levels at T1 in normal-weight participants with a positive COVID-19 disease history before immunization divided by age. T1: 14 days after BNT162b2 booster dose, AB: antibodies, OD: optical density.

**Table 1 diseases-13-00078-t001:** Anti-SARS-CoV-2 IgG OD ratio and SARS-CoV-2 neutralizing AB (%) divided by BMI in immune-naïve and convalescent subgroups. BMI: body mass index, OD: optical density, AB: antibody, IgG: immunoglobulin G. T0: prior to vaccination, T1: 14 days after BNT162b2 booster dose, T2: 6 months after BNT162b2 booster dose.

	Number	Anti-SARS-CoV-2 IgG ELISA OD Ratio		SARS-CoV-2 Neutralizing AB (%)	
Sampling		T0 Mean (95%CI)	T1 Mean (95%CI)	T0 Median (IQR)	T1 Median (IQR)
Negative disease history prior to immunization					
BMI					
Normalweight	26	0.09 (0.06–0.12)	2.55 (2.50–2.60)	2.60 (1.67)	97.51 (2.22)
Overweight–obese	20	0.08 (0.45–0.12)	2.42 (2.31–2.52)	2.43 (1.23)	96.34 (2.28)
*p*-value		0.599	0.028	0.871	0.008
Positive disease history prior to immunization					
BMI					
Normalweight	17	1.50 (1.22–1.78)	2.73 (2.66–2.80)	50.92 (44.16)	97.8 (1.55)
Overweight–obese	16	1.07 (0.79–1.36)	2.57 (2.44–2.70)	15.45 (36.86)	96.93 (3.96)
*p*-value		0.026	0.032	<0.001	0.067

**Table 2 diseases-13-00078-t002:** Demographic characteristics, lifestyle parameters (smoke exposure, physical activity, KIDMED score) and laboratory findings of normal-weight and overweight–obese participants. BMI: body mass index. WBC: white blood cells, RBC: red blood cells, HGB: hemoglobulin, PLT: platelets, ESR: erythrocyte sedimentation rate, TCHOL: total cholesterol, TG: triglycerides, LDL: low-density lipoprotein, HDL: high-density lipoprotein, Lp (a): lipoprotein (a), CRP: C-reactive protein, HBA1C: hemoglobin A1C, CI: confidence interval, IQR: interquartile range, FGIR: fasting serum glucose (mg/dL) to plasma insulin (micro U/mL) ratio. Body mass index (BMI) defined as weight (kg)/height^2^ (m^2^) according to Center of Disease Control (CDC) curves for age and gender.

			BMI Group		*p*-Value
			Normal-Weight	Overweight–Obese	
Gender	Female	Number (%)	22 (55)	18 (45)	0.921
Age (years)	5–11	Number (%)	15 (51.7)	14 (48.3)	0.713
	12–18	Number (%)	28 (56)	22 (44)	
Tanner I-II	Female	Number (%)	8 (50)	8 (50)	0.604
Tanner III-V	Female	Number (%)	14 (58.3)	10 (41.7)	
Tanner I-II	Male	Number (%)	7 (53.8)	6 (46.2)	0.987
Tanner III-V	Male	Number (%)	14 (53.8)	12 (46.2)	
Smoking Score ^a^	1	Number (%)	29 (72.5)	11 (27.5)	<0.001
	0	Number (%)	11 (28.2)	28 (71.8)	
Exercise Score ^b^	0	Number (%)	14 (43.8)	18 (56.3)	0.212
	1	Number (%)	29 (61.7)	18 (38.3)	
Kidmed Score ^c^	Poor (0–3)	Number (%)	6 (50)	6 (50)	0.751
	Moderate (4–7)	Number (%)	24 (58.5)	41 (41.5)	
	Good (≥8)	Number (%)	13 (50)	13 (50)	
Hematological parameters	WBC (10^3^/μL) median (IQR)		6.70 (3)	7.55 (2.75)	0.509
	RBC (10^6^/μL) median (IQR)		4.58 (0.47)	4.73 (0.46)	0.817
	HGB (g/dL) mean (95% CI)		13.2 (12.9–13.5)	13.4 (12.9–13.8)	0.429
	PLT (10^3^/μL) mean (95% CI)		261 (236–286)	271 (246–296)	0.574
	ESR (mm/hr) median (IQR)		8 (5)	8 (6)	0.350
Biochemical parameters	TCHOL (mg/dL) mean (95% CI) ^d^		150.19 (144.65–155.72)	150.23 (142.41–158.05)	0.924
	TG (mg/dL) mean rank ^d^		33.73	36.59	0.135
	LDL (mg/dL) mean (95% CI) ^d^		81.82 (76.54–87.11)	88.89 (81.57–96.20)	0.269
	HDL (mg/dL) mean (95% CI) ^d^		61.60 (57.87–65.34)	54.29 (50.13–58.44)	0.006
	Lp(a) (mg/dL) mean rank ^d^		38.23	41.06	0.581
	CRP (mg/l) mean rank ^e^		30.79	51.00	<0.001
	HBA1c (%) mean rank ^e^		33.36	46.67	0.006
	Uric acid (mg/dL) mean (95% CI) ^f^		4.19 (3.91–4.47)	4.85 (4.42–5.28)	0.009
	FGIR mean rank ^f^		50.51	26.65	<0.001

^a^ Secondhand smoking was considered significant when the legal guardians were active indoor smokers with more than 20 pack years. ^b^ According to Center of Disease Control (CDC) physical activity key guidelines for children and adolescents, exercise should include vigorous-intensity, muscle- and bone-strengthening physical activity at least three times per week. ^c^ According to KIDMED index, a score of 0–3 reflects poor adherence to the Mediterranean diet, a score of 4–7 describes average adherence, and a score of 8–12 good adherence. ^d^ National Cholesterol Education Program (NCEP) Definition for Dyslipidemia in Children and Adolescents abnormal values were as follows: TCHOL ≥200 mg/dL, TGs ≥100 mg/dL for ≤9 years old and ≥130 mg/dL for >10 years old, LDL-C ≥130 mg/dL, HDL-C <40 mg/dL, and Lp (a) ≥30. ^e^ Normal ranges of CRP: 0–5 mg/L, HBA1c: 4.5–6.3% according to the laboratory manufacturer design. ^f^ According to the literature, values were considered abnormal when FGIR <6.5 both for prepubertal and pubertal participants and uric acid were >6.5 mg/dL.

**Table 3 diseases-13-00078-t003:** Stepwise method of multivariate linear regression of anti-SARS-CoV-2 IgG ELISA OD ratio and SARS-CoV-2 neutralizing AB levels with BMI, age, gender, biochemical profile (uric acid, TG, FGIR, and HBA1c) and with lifestyle parameters (smoke exposure, physical activity, KIDMED score) and according to their COVID-19 disease history prior to vaccination both at T0 and T1 (T0: before vaccination, T1: 14 days after BNT162b2 booster dose). BMI: body mass index, FGIR: fasting serum glucose (mg/dL) to plasma insulin (micro U/mL) ratio, HBA1c: hemoglobin A1c, TG: triglycerides, R^2^: adjusted R square.

		COVID-19 Disease History Prior to Vaccination				
Dependent Variables	Independent Variables	Negative		Independent Variables	Positive	
		Beta (95% CI)	*p*-Value		Beta (95% CI)	*p*-Value
T0 anti-SARS-CoV-2 IgG	-	-	-	Smoke exposure	−0.55 (−0.91;−0.20)	0.004 (R^2^: 0.22)
Τ0 SARS-CoV-2 Neutralizing (%)	-	-	-	Smoke exposure	−25.61(−42.77;−8.44)	0.005 (R^2^: 0.21)
T1 anti-SARS-CoV-2 IgG	BMI	−0.02 (−0.03; −0.01)	0.002 (R^2^: 0.19)	BMI	−0.03 (−0.04; −0.02)	<0.001 (R^2^: 0.45)
	Uric acid	−0.06 (−0.114; −0.011)	0.018 (R^2^: 0.11)	FGIR	−0.24 (−0.38; −0.11)	0.001 (R^2^: 0.44)
				HBA1c	−0.17 (−0.33; −0.02)	0.027 (R^2^: 0.44)
Τ1 SARS-CoV-2 Neutralizing (%)	BMI	−0.17 (−0.25; −0.11)	<0.001 (R^2^: 0.30)	BMI	−0.15 (−0.26; −0.04)	0.009 (R^2^: 0.30)
				FGIR	−1.52 (−2.62; −0.42)	0.008 (R^2^: 0.35)
	Uric acid	−0.59 (−0.95; −0.23)	0.002 (R^2^: 0.19)	HBA1c	−1.36 (−2.63; −0.10)	0.038 (R^2^: 0.35)

## Data Availability

All data are available from the corresponding author upon reasonable request.

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
