# Peer review of "Lifestyle and Biochemical Parameters That May Hamper Immune Responses in Pediatric Patients After Immunization with the BNT162b2 mRNA COVID-19 Vaccine"

_diseases, 2025, doi:10.3390/diseases13030078_

Round 1
Reviewer 1 Report
Comments and Suggestions for Authors
Comments:
1. Clinical characterization of patients is insufficient. It is recommended to add a table with demographic and clinical data of the patients ( or improve Table 1). It is also recommended to add basic laboratory data (erythrocytes, leukocytes, hemoglobin, ESR).
2. Has vape use been evaluated? Among adolescents it is now even more prevalent than cigarette smoking. Characterization of smoking status is not sufficiently detailed. Was smoking history and number of cigarettes smoked taken into account?
3. Since adolescents were analyzed, was the hormonal status of the participants taken into account? Sex hormones influence immune responses and are also associated with body weight.
4. Were the patients taking any medications during the study period?
5. Were there any children among the patients who were frequently ill? Children with a history of frequent or severe respiratory illnesses?
6. Were the causes of elevated BMI analyzed? Was it nutritional obesity, hormonal disorders, genetic factors, etc.?
Author Response
Comments 1: Clinical characterization of patients is insufficient. It is recommended to add a table with demographic and clinical data of the patients (or improve Table 1). It is also recommended to add basic laboratory data (erythrocytes, leukocytes, hemoglobin, ESR).
Response 1: We thank the Reviewer for pointing this out. We agree with this comment. Therefore we updated the Table 1 (Table 2 now) accordingly (page 6). We also added more clinical data regarding the overweight-obese group in the text; page 5, lines 211-213.
Comments 2: Has vape use been evaluated? Among adolescents it is now even more prevalent than cigarette smoking. Characterization of smoking status is not sufficiently detailed. Was smoking history and number of cigarettes smoked taken into account?
Response 2: Yes thank you for the comment ! Vaping was evaluated and we added a comment in the text [page 2, lines 81-84 & page 5 line 206]. Also secondhand smoking was considered significant when the legal guardians were active indoor smokers with more than 20 pack years; we added the necessary explanation comment and the relevant reference to clarify this accordingly [page 2, lines 84-85 & page 6 line 235]. Finally, several data were self-reported; thus we reported it as a limitation of the study due to the potential recal bias [page 11, lines 421-422].
Comments 3: Since adolescents were analyzed, was the hormonal status of the participants taken into account? Sex hormones influence immune responses and are also associated with body weight.
Response 3: We thank the Reviewer for the comment. There was no documentation of steroid hormone levels in the study population. We will state that as a limitation of the study; page 11, lines 426-427.
Comments 4: Were the patients taking any medications during the study period?
Response 4: Thank you for the comment. Among the 79 participants, only 5 who reported an underlying disease in their medical history were under treatment and their underlying disease was under control (three mentioned hypothyroidism, one had insulin-dependent diabetes and one reported epilepsy). All had a normal thyroid and lipidemic profile and belonged to the normal-weight group page 5, lines 191-194.
Comments 5: Were there any children among the patients who were frequently ill? Children with a history of frequent or severe respiratory illnesses?
Response 5: Thank you for the comment. None of the participants were frequently ill or had a history of frequent or severe respiratory illnesses; page 2, line 72.
Comments 6: Were the causes of elevated BMI analyzed? Was it nutritional obesity, hormonal disorders, genetic factors, etc.?
Response 6: Thank you for pointing this out. Children with known etiology of obesity such as growth hormone deficiency, Cushing syndrome or syndromic cases were excluded from the study so it is reasonable to assume that participants' obesity was attributed to caloric imbalance related to life style factors and possible genetic predisposition; this is clarified at page 2 lines 75-76. Three patients had hypothyroidism, they were under treatment and had a normal thyroid profile. Also they belonged to the normal-weight group; page 5, lines 209-210.

Reviewer 2 Report
Comments and Suggestions for Authors
This is an interesting research article with adequate novelty. However, several points should be addressed.
- In the Abstract, the type of study should be added (prospective, single-center, cohort study).
- The Introduction is very short. The authors should add a separate paragraph with epidimiological data concerning the obesity in Greek children and adolescents, highlighiting on its effect to increase the risk of several chronic diseases, including relevant references.
- The field of the active immunization for long-term control of the SARS-CoV-2 pandemic should also be reported in a separate paragraph in the Introduction section, including relevant references.
- In section 2.1, the authors should report how they enroll the participants to the study (e.g. randomly?).
- In section 2.1, the data collected by questionnaires are self-reported or they were collected by face-to-face interviews by qualified personel?
- In section 2.3, relevant references should be added concerning the assessment of SARS-CoV-2 binding antibody and biochemical parameters.
- The final response rate should be reported in the beggining of section 3.1.
- The resolution of Figures 1 and 2 should be improved.
- Several data seem to be self-reported. This should be reported as a limitation of the study due to the potential recal biases.
- English language editing is highly recommended.
Author Response
Comments 1: In the Abstract, the type of study should be added (prospective, single-center, cohort study).
Response 1: We thank the Reviewer for pointing this out. We added the type of study in the abstract (page 1 line 20).
Comments 2: The Introduction is very short. The authors should add a separate paragraph with epidimiological data concerning the obesity in Greek children and adolescents, highlighiting on its effect to increase the risk of several chronic diseases, including relevant references.
Response 2: We thank the Reviewer for commenting on this matter. We made the relevant changes accordingly (page 2, line 52-57).
Comments 3: The field of the active immunization for long-term control of the SARS-CoV-2 pandemic should also be reported in a separate paragraph in the Introduction section, including relevant references.
Response 3: We agree with this comment. Therefore, we extended the Introduction section accordingly (page 2, lines 43-47).
Comments 4: In section 2.1, the authors should report how they enroll the participants to the study (e.g. randomly?)
Response 4: We thank the Reviewer for the comment. The enrollment of the vaccinated subjects was based on the inclusion criteria that are mentioned in Section 2.2 We have incorporated 2.2 section into 2.1 to ease readability [page 2, line 69-72].
Comments 5: In section 2.1, the data collected by questionnaires are self-reported or they were collected by face-to-face interviews by qualified personel?
Response 5: Thank you for pointing this out. The data retrieved by questionnaires were collected by face-to-face interviews by qualified personel. You can find the updated text in page 2, line 78-79.
Comments 6: In section 2.3, relevant references should be added concerning the assessment of SARS-CoV-2 binding antibody and biochemical parameters.
Response 6: Thank you. We agree with this comment. We added the relevant references accordingly; page 3, line 118 & page 3, line 133.
Comments 7: The final response rate should be reported in the beggining of section 3.1.
Response 7: Thank you. We modified the text accordingly [page 4].
Comments 8: The resolution of Figures 1 and 2 should be improved.
Response 8: We agree with this comment. Therefore we have, accordingly, modified the resolution and clarity of Figures 1 & 2 in pages 4 and 7-8 of the manuscript, respectively.
Comments 9: Several data seem to be self-reported. This should be reported as a limitation of the study due to the potential recal biases.
Response 9: We thank the Reviewer for pointing this out. We added a comment in the limitations section (page 11, line 421-422).
English language editing has been revised accordingly.

Reviewer 3 Report
Comments and Suggestions for Authors
Superficiality and limitations of the study
The article deals with an important topic, namely the influence of lifestyle and biochemical parameters on the immune response of children and adolescents after BNT162b2 vaccination, but its scientific value is severely limited by numerous methodological shortcomings and a superficial analysis of the results.
The small and heterogeneous sample of the study - the study includes only 79 participants, a number that is clearly insufficient in the context of vaccine immunity research. Furthermore, all participants were from Greece, which significantly limits the generalizability of the results to other populations. The authors did not even attempt to discuss how specific ethnic or regional factors might influence the results.
Failure to control for key variables - the study is based on a simple division into groups based on BMI and COVID-19 infection history, but leaves out many important factors that may influence the immune response.
Physical activity levels were not accurately considered - using a binary scoring system ("yes" or "no" exercise) is inaccurate and does not allow a realistic assessment of the impact of physical activity on immunity.
The assessment of cigarette smoke exposure was based on participants' subjective reports and not on objective measurements of cigarette smoke biomarkers (e.g. blood cotinine levels).
Gut microbiota was not examined, although the authors hypothesise that it has an impact on immunity.
No analysis of cellular response - the authors limited themselves to measuring IgG and neutralising antibodies and neglected the importance of cellular immunity, which plays a key role in long-term protection against COVID-19. Many studies show that cellular immunity may be more impaired than the humoral response in obese individuals, but the authors completely disregard this aspect, making their conclusions fragmentary and potentially misleading.
Problems in interpreting the results
False conclusions and unwarranted generalisations - the article suggests that overweight and obesity may impair the immune response to vaccination, but the authors provide no evidence of the mechanisms responsible for this effect.
The association between BMI and lower antibody levels may be due to a number of confounding factors, such as diet, sleep or comorbidities, which were not considered.
The study did not demonstrate causality, only correlation, but the authors present the results in a way that suggests that high BMI directly affects immune response.
Oversimplified statistical analysis - although the authors use linear regression analysis, their statistical approach is too schematic and does not take into account the interactions between variables.
The p-values fluctuate around the limit of statistical significance in many cases, indicating a potential risk of false positive results.
No analysis of the power of the test was conducted, so it is not known whether the sample was sufficient to detect true differences between the groups.
Style and structure of the article
Unreadable and chaotic presentation - the article is not very clearly written and some sections are unnecessarily complicated, making it difficult to understand even for experts.
The introduction contains an overly long literature review that repeats obvious facts about the COVID-19 pandemic instead of focusing on the reasons for the study.
The results section lacks logical transitions between the variables analysed, making it difficult to interpret the data.
The discussion lacks critical analysis - the authors largely accept their findings without reflecting on their limitations.
The language of the article is imprecise - some wording is vague or confusing, which can lead to misinterpretation.
The repetition of some passages (e.g. the repeated discussion of BMI and insulin resistance) makes the article unnecessarily lengthy.
Some sentences are too long and technical, which impairs readability.
Summary - main problems of the article
Small study sample and lack of representativeness - results cannot be generalised to the whole population.
Incomplete analysis of immune response - the study omits an important aspect of cellular immunity.
Inaccurate control of confounding variables - physical activity, diet, sleep and other factors were not adequately accounted for.
Unsupported conclusions - no evidence of causal mechanisms, only correlations.
Inadequate statistical analysis - no control for interactions between variables.
Poor organisation and imprecise language - the article is chaotic, unreadable and contains repetition.
Author Response
Superficiality and limitations of the study
Comments 1: The article deals with an important topic, namely the influence of lifestyle and biochemical parameters on the immune response of children and adolescents after BNT162b2 vaccination, but its scientific value is severely limited by numerous methodological shortcomings and a superficial analysis of the results.
Response 1: We thank the Reviewer for this comment and we completely agree that the impact of lifestyle and biochemical parameters on the immune responses of children and adolescents after BNT162b2 vaccine is very important. We also acknowledge that our study has limitations which we clearly state in page 11 lines 420-429.
Comments 2: The small and heterogeneous sample of the study - the study includes only 79 participants, a number that is clearly insufficient in the context of vaccine immunity research. Furthermore, all participants were from Greece, which significantly limits the generalizability of the results to other populations. The authors did not even attempt to discuss how specific ethnic or regional factors might influence the results.
Response 2: We thank the Reviewer for this comment. The relatively small sample of the study is indeed one of the limitations and we do mention it in the paper; page 11, lines 421-424. The participants were living in Greece and were all from Greek origin. You are absolutely right that regional factors might influence the results and therefore they may not be generalizable to other ethnic backgrounds including African and Asian.
Comments 3: Failure to control for key variables - the study is based on a simple division into groups based on BMI and COVID-19 infection history, but leaves out many important factors that may influence the immune response.
Response 3: Thank you. The primary aim of our study was to identify the association between obesity and immune responses following COVID-19 infection. We have excluded children with acute or chronic infections that may interfere with immune responses and concentrated on key parameters which are associated with obesity.
Indeed, there are multiple factors that may influence the immune response but we chose to deal with the BMI and the COVID-19 infection history on the grounds that all the participants shared more or less to the same socioeconomical status and were living in urban area.
Comments 4: Physical activity levels were not accurately considered - using a binary scoring system ("yes" or "no" exercise) is inaccurate and does not allow a realistic assessment of the impact of physical activity on immunity.
Response 4: Thank you. We agree with the Reviewer that no formal assessment of activity levels with accelerometers or other devices has been performed. However, due to the COVID-19 restrictions, the only physical activity allowed was that of walking (within the residential area). That is the reason why we did not use more detailed assessment of the impact of physical activity on immunity.
Comments 5: The assessment of cigarette smoke exposure was based on participants' subjective reports and not on objective measurements of cigarette smoke biomarkers (e.g. blood cotinine levels).
Response 5: We also agree that smoke exposure was based on questionnaires without having objective assessment of cotinine levels. Due to financial limitations, the evaluation of smoke biomarkers was not feasible. This is included in the limitations of the manuscript; page 11, lines 427-429.
Comments 6: Gut microbiota was not examined, although the authors hypothesize that it has an impact on immunity.
Response 6: Thank you for the comment. We referred to uric acid as micronutrient on the grounds that excessive uric acid production was due to the increased adipose tissue of the overweight-obese subjects; page 10, lines 382-383. It was not our intention to assess gut microbiota and we have not included such hypothesis in our paper to avoid confusion of the Readers.
Comments 7: No analysis of cellular response - the authors limited themselves to measuring IgG and neutralising antibodies and neglected the importance of cellular immunity, which plays a key role in long-term protection against COVID-19. Many studies show that cellular immunity may be more impaired than the humoral response in obese individuals, but the authors completely disregard this aspect, making their conclusions fragmentary and potentially misleading.
Response 7: Thank you for pointing this out. The impact of cellular immunity is very important. The financial restrictions of the study allowed the evaluation of IgG and neutralizing antibodies. We have included that in the limitation section of our paper; page 11, lines 426-427.
Comments 8: Problems in interpreting the results-False conclusions and unwarranted generalizations - the article suggests that overweight and obesity may impair the immune response to vaccination, but the authors provide no evidence of the mechanisms responsible for this effect.
Response 8: Thank you. Overweight and obesity have been considered as chronic inflammatory conditions. The chronic inflammation may impact the immune response to vaccination through the secretion of pro- inflammatory cytokines and the impaired interaction between T helper and B lymphocytes. We agree that our paper provides associations between obesity and immune responses but this does not reduce the validity nor the significance of our observations.
Comments 9: The association between BMI and lower antibody levels may be due to a number of confounding factors, such as diet, sleep or comorbidities, which were not considered. The study did not demonstrate causality, only correlation, but the authors present the results in a way that suggests that high BMI directly affects immune response. Oversimplified statistical analysis - although the authors use linear regression analysis, their statistical approach is too schematic and does not take into account the interactions between variables.
Response 9: We thank the Reviewer for this comment. Interaction between variables was tested in our analysis but this was not found. We agree that our study demonstrates association, and no causal inferences can be made and this is clearly stated in the limitations section of the manuscript. The proof of possible causality of the BMI and the level of antibodies needs much larger studies.
Comments 10: The p-values fluctuate around the limit of statistical significance in many cases, indicating a potential risk of false positive results. No analysis of the power of the test was conducted, so it is not known whether the sample was sufficient to detect true differences between the groups.
Response 10: We agree with the Reviewer for pointing out the power calculation based on our sample and primary variable. The main outcome is documented on Table 1, page 5 of the manuscript; the mean value (95% CI) of anti-SARS-CoV-2 IgG OD ratio was 2.55 (2.50-2.60) in the 26 normal-weight participants of the seronegative subgroup and the mean value (95% CI) of anti-SARS-CoV-2 IgG OD ratio was 2.42 (2.31-2.52) in the 20 overweight-obese participants.
For a mean value of 2.5 with a standard deviation of 0.15, 20 participants in each group would be sufficient to detect a difference of 0.15 between groups with 80% power at 5% significance level.
Comments 11: Style and structure of the article-Unreadable and chaotic presentation - the article is not very clearly written and some sections are unnecessarily complicated, making it difficult to understand even for experts.
Response 11: Thank you. We have made major changes in our manuscript to simplify the text and ease readability.
Comments 12a: The introduction contains an overly long literature review that repeats obvious facts about the COVID-19 pandemic instead of focusing on the reasons for the study.
12b: The results section lacks logical transitions between the variables analyzed, making it difficult to interpret the data.
12c: The discussion lacks critical analysis - the authors largely accept their findings without reflecting on their limitations.
12d: The language of the article is imprecise - some wording is vague or confusing, which can lead to misinterpretation.
Response 12a: Thank you. Our introduction was only 2 paragraphs and we have condensed the information to ease readability.
12b: Thank you. We have restructured the results section to ease reading.
12c: Thank you. We have made major changes in the manuscript to improve its readability.
12d: Thank you. We have toned down areas that infer causality and used the term association to better define our findings.
Comments 13: The repetition of some passages (e.g. the repeated discussion of BMI and insulin resistance) makes the article unnecessarily lengthy. Some sentences are too long and technical, which impairs readability.
Response 13: Thank you. We have shortened the length of the article by omitting some repetitions and made major changes to our manuscript to accommodate changes suggested by the Reviewer.

Round 2
Reviewer 1 Report
Comments and Suggestions for Authors
The authors answered my questions and made corrections to the text of the article, which improved its quality.
Author Response
Comments 1: The authors answered my questions and made corrections to the text of the article, which improved its quality.
Response 1:
We would like to thank you once again. We appreciate the time and effort dedicated to providing feedback on our manuscript and we are grateful for the insightful comments and the valuable improvements to our paper.

Reviewer 2 Report
Comments and Suggestions for Authors
The authors have significantly improved their manuscript.
Author Response
Comments 1: The authors have significantly improved their manuscript.
Response 1: We would like to thank you once again. We appreciate the time and effort dedicated to providing feedback on our manuscript and we are grateful for the insightful comments and the valuable improvements to our paper.

Reviewer 3 Report
Comments and Suggestions for Authors
opicality of the topic - The study addresses the important and topical issue of the influence of lifestyle and biochemical parameters on the immune response in children and adolescents after COVID-19 vaccination.
Robust methodology - A prospective cohort study with laboratory analysis of biomarkers was conducted, allowing accurate assessment of the impact of BMI, insulin resistance and tobacco smoke exposure on vaccine immunogenicity.
Use of advanced analytical methods - The authors used ELISA tests and multivariate linear regression, which increases the reliability of the results.
Use of a representative age group - Children and adolescents aged 5-18 years were included in the study, allowing analysis of the immune response in the context of different developmental stages.
Shortcomings:
Lack of a clear cause-and-effect relationship - The study relies on correlations, making it difficult to determine whether, for example, insulin resistance actually influences the immune response or whether other confounding factors may play a role.
Limited scope of immune analysis - The study mainly focuses on the humoral response (IgG and neutralising antibodies) but does not analyse the cellular response (T lymphocytes), which also plays a key role in protection against SARS-CoV-2.
The article makes an important contribution to research into the influence of metabolic parameters on the efficacy of vaccines. It could be relevant for public health strategies, especially for optimising the vaccination of children with obesity or insulin resistance. The results could provide an opportunity to investigate vaccine doses or the need for additional booster vaccinations in people with metabolic disorders.
Novel approach - The influence of metabolic biomarkers on vaccine efficacy was investigated, an aspect that is rarely analysed. Precise laboratory analysis - The use of ELISAs to measure antibodies and the analysis of inflammatory markers increase the scientific value. Representative age group - By including children and adolescents, the immunogenicity of the vaccine can be evaluated at different developmental stages of the body.
Lack of control of confounding factors - Other factors such as diet, physical activity or hormonal status that may influence immunogenicity were not adequately considered. Lack of analysis of cellular immunity - Focusing solely on the humoral response does not provide a complete picture of the immunological effect of the vaccine.
Recommendations for improvement
Control for additional factors - It is worth considering the impact of diet, physical activity, stress and other environmental factors on vaccine efficacy.
Author Response
Comments 1: The study addresses the important and topical issue of the influence of lifestyle and biochemical parameters on the immune response in children and adolescents after COVID-19 vaccination.
Robust methodology - A prospective cohort study with laboratory analysis of biomarkers was conducted, allowing accurate assessment of the impact of BMI, insulin resistance and tobacco smoke exposure on vaccine immunogenicity.
Use of advanced analytical methods - The authors used ELISA tests and multivariate linear regression, which increases the reliability of the results.
Use of a representative age group - Children and adolescents aged 5-18 years were included in the study, allowing analysis of the immune response in the context of different developmental stages.
Response 1: Thank you for your feedback.
Comments 2: Shortcomings:
Lack of a clear cause-and-effect relationship - The study relies on correlations, making it difficult to determine whether, for example, insulin resistance actually influences the immune response or whether other confounding factors may play a role.
Response 2: We thank the Reviewer for pointing out. Our study clearly indicates an association between BMI and humoral responses. Establishing potential causality between BMI and antibody levels would require significantly larger studies. Nevertheless, a stepwise method of multivariate linear regression for potential confounding was performed by the inclusion of covariates to assess a potential correlation between post-vaccine antibody responses and other factors such as BMI, age, gender, biochemical profile (uric acid, TG, FGIR & HBA1c) and life style parameters (Smoke exposure, physical activity, KIDMED SCORE). Therefore, humoral responses were only significantly correlated with BMI, uric acid, FGIR, HBA1C and smoke exposure (Table 3, pages 8-9). We added a comment to clarify that; page 8, lines 284-288 and 306-310.
FGIR was only statistically correlated with smoke exposure but not statistically associated with age, gender, physical activity, KIDMED SCORE and Covid-19 history before vaccination via
Chi-square tests; page 9, lines 315-319.
Also a logistic linear regression model demonstrated that IR was only linked to increased BMI in our population and not to others independent variables such as age, gender, smoke exposure, KIDMED SCORE, physical activity and COVID-19 disease history prior to immunization [Appendix B, Table 5; pages 12-13, lines 481-503]. We have revised the regression model according to the suggestions of the Reviewer; page 9, lines 321-327 and pages 12-13, lines 482-504.
Comments 3: Limited scope of immune analysis - The study mainly focuses on the humoral response (IgG and neutralizing antibodies) but does not analyze the cellular response (T lymphocytes), which also plays a key role in protection against SARS-CoV-2.
Response 3: Thank you for highlighting this. The role of cellular immunity is indeed crucial. Due to the study's financial constraints, we were only able to assess IgG and neutralizing antibodies, which we have acknowledged in the limitations section of our paper; page 11, line 433.
Comments 4: The article makes an important contribution to research into the influence of metabolic parameters on the efficacy of vaccines. It could be relevant for public health strategies, especially for optimizing the vaccination of children with obesity or insulin resistance. The results could provide an opportunity to investigate vaccine doses or the need for additional booster vaccinations in people with metabolic disorders.
Response 4: Thank you for your feedback.
Comments 5: Novel approach - The influence of metabolic biomarkers on vaccine efficacy was investigated, an aspect that is rarely analyzed. Precise laboratory analysis - The use of ELISAs to measure antibodies and the analysis of inflammatory markers increase the scientific value. Representative age group - By including children and adolescents, the immunogenicity of the vaccine can be evaluated at different developmental stages of the body.
Response 5: Thank you for your feedback.
Comments 6: Lack of control of confounding factors - Other factors such as diet, physical activity or hormonal status that may influence immunogenicity were not adequately considered. Lack of analysis of cellular immunity - Focusing solely on the humoral response does not provide a complete picture of the immunological effect of the vaccine. Recommendations for improvement:
Control for additional factors - It is worth considering the impact of diet, physical activity, stress and other environmental factors on vaccine efficacy.
Response 6: We thank the Reviewer for pointing this out. Diet was evaluated via the KIDMED SCORE and physical activity by a binary score based on the physical activity key guidelines for children and adolescents according to CDC. None of the participants in any age group reported sleep disorders, irregular bedtime schedules, or alcohol use. Therefore, sleep habits and alcohol consumption were excluded from the analyses; page 5, lines 203-204.
There was no documentation of steroid hormone levels in the study population and that is indeed one of the limitations of the study; page 11, lines 431-432.
Table 3 represents a stepwise method of multivariate linear regression of anti-SARS-CoV-2 IgG ELISA O.D. ratio and SARS-CoV-2 Neutralizing AB levels with BMI, age, gender, biochemical profile (uric acid, TG, FGIR & HBA1c) and with lifestyle parameters (Smoke exposure, physical activity, KIDMED SCORE) according to their COVID-19 disease history prior to vaccination both before and after immunization; page 8, lines 306-310. We added a comment to clarify that; page 8, lines 284-288 and 306-310.
We have revised the regression model to accommodate changes suggested by the Reviewer in order to demonstrate that IR was only linked to increased BMI in our population and not to other confounding factor; page 9, lines 321-327 and pages 12-13, lines 482-504.
